# Endometriosis and the Role of Pro-Inflammatory and Anti-Inflammatory Cytokines in Pathophysiology: A Narrative Review of the Literature

**DOI:** 10.3390/diagnostics14030312

**Published:** 2024-01-31

**Authors:** Ioan Emilian Oală, Melinda-Ildiko Mitranovici, Diana Maria Chiorean, Traian Irimia, Andrada Ioana Crișan, Ioana Marta Melinte, Teodora Cotruș, Vlad Tudorache, Liviu Moraru, Raluca Moraru, Laura Caravia, Mihai Morariu, Lucian Pușcașiu

**Affiliations:** 1Department of Obstetrics and Gynecology, Emergency County Hospital Hunedoara, 331057 Hunedoara, Romania; oalaioanemilian@gmail.com; 2Department of Pathology, County Clinical Hospital of Targu Mures, 540072 Targu Mures, Romania; chioreandianamaria@yahoo.com; 3Doctoral School of Medicine and Pharmacy, George Emil Palade University of Medicine, Pharmacy, Science, and Technology of Targu Mures, 540142 Targu Mures, Romania; drtraianirimia@gmail.com (T.I.); crisanandrada@yahoo.com (A.I.C.); ioanammelinte@gmail.com (I.M.M.); cotrus.teodora12@gmail.com (T.C.); 4Department of 1st Gynecology Clinic, Emergency County Hospital Targu Mures, 540136 Targu Mures, Romania; 5Department of 2nd Gynecology Clinic, County Clinical Hospital Targu Mures, 540072 Targu Mures, Romania; vladtudorache1994@gmail.com; 6Department of Anatomy, George Emil Palade University of Medicine, Pharmacy, Sciences and Technology, 540142 Targu Mures, Romania; liviu.moraru@umfst.ro (L.M.); raluca.moraru@umfst.ro (R.M.); 7Department of Morphological Sciences, Division of Cellular and Molecular Biology and Histology, Carol Davila University of Medicine and Pharmacy, 050474 Bucharest, Romania; laura.caravia@umfcd.ro; 8Department of Obstretics and Gynecology, George Emil Palade University of Medicine and Pharmacies, Science and Technology of Targu Mures, 540142 Targu Mures, Romania; mmorariu@gmail.com (M.M.); puscasiu@gmail.com (L.P.)

**Keywords:** endometriosis, ectopic endometrial tissue, pro-inflammatory cytokines, pathogenic mechanism, anti-inflammatory cytokines

## Abstract

Endometriosis is a chronic inflammatory disease, which explains the pain that such patients report. Currently, we are faced with ineffective, non-invasive diagnostic methods and treatments that come with multiple side effects and high recurrence rates for both the disease and pain. These are the reasons why we are exploring the possibility of the involvement of pro-inflammatory and anti-inflammatory molecules in the process of the appearance of endometriosis. Cytokines play an important role in the progression of endometriosis, influencing cell proliferation and differentiation. Pro-inflammatory molecules are found in intrafollicular fluid. They have an impact on the number of mature and optimal-quality oocytes. Endometriosis affects fertility, and the involvement of endometriosis in embryo transfer during in vitro fertilization (IVF) is being investigated in several studies. Furthermore, the reciprocal influence between anti-inflammatory and pro-inflammatory cytokines and their role in the pathogenesis of endometriosis has been assessed. Today, we can affirm that pro-inflammatory and anti-inflammatory cytokines play roles in survival, growth, differentiation, invasion, angiogenesis, and immune escape, which provides a perspective for approaching future clinical implications and can be used as biomarkers or therapy.

## 1. Introduction

Endometriosis is a pathology characterized by ectopic endometrial tissue implanted and developed on host tissues [1]. It is characterized by pain, dysmenorrhea, dyspareunia, infertility, and pelvic organ dysfunction [2,3]. Endometriosis affects fertility, and the involvement of endometriosis in embryo transfer during in vitro fertilization (IVF) is being investigated. It appears that inflammation plays a key role in the failure of embryo transfer, but we currently lack clear data on how it affects pregnancy outcomes [4].

Currently, we are faced with ineffective, non-invasive diagnostic methods and treatments that come with multiple side effects and high recurrence rates of both the disease and pain [2]. Moreover, the disease is characterized by an impaired quality of life, a negative impact on social and family life, mental health issues, and high healthcare costs [2]. Hence, there is great attention and importance given to this pathology, and there is an inclination towards understanding its etiopathogenesis as the main starting point for finding new diagnostic and treatment methods.

Similar to cancer, ectopic endometrial tissue can lead to tissue spread, invasion, organ damage, and neoangiogenesis. It has been established that a history of endometriosis is more common in women with invasive clear cell tumors and endometrioid neoplasms compared to healthy women. Some cytokines show increased levels in both pathologies, indicating a link between cytokine levels and the pathophysiology of endometriosis. Cytokines play an important role in the progression of endometriosis, influencing cell proliferation and differentiation. Endometriosis may serve as a precursor for ovarian cancer [5].

The pathophysiology of endometriosis is not fully understood. There is a combination of causes, including hormonal, immunological, genetic (gene polymorphism), and environmental factors [2,3]. Theories on the histogenesis of endometriosis include retrograde menstruation, celomic metaplasia, embryogenic cell rests, induction, and lymphatic and vascular dissemination. Genetic alterations, elevated gonadotropins, elevated estrogens, progesterone deficiency, and chronic inflammation are all implicated in the development of endometriosis [6].

The most widely accepted explanation of the etiopathogenic mechanism is retrograde menstruation, with endometrial epithelial and stromal cells implanting in the peritoneal cavity. The lesions are classified based on the following colors: white, red, brown, and black, with the latter being due to the presence of glandular content and adjacent stromal reactions rather than the severity of the disease, an important detail for the histopathologist [1]. Genetic and epigenetic factors, as well as environmental factors, are essential because 90% of women experience retrograde menstruation, but only a few develop endometriosis. Additionally, hormonal factors play a role, as elevated estrogen levels lead to bleeding in ectopic lesions, with the secondary release of pro-inflammatory cytokines that contribute to iron overload. This results in the infiltration of monocytes and macrophages, which stimulate lipid peroxidation and the accumulation of malondialdehyde (MDA) in the stroma [1]. Thus, there is a link between reactive oxygen species (ROS) and pro-inflammatory factors that contribute to pain and the failure to detoxify lipid peroxidase products under oxidative stress. IL-1 beta, IL-6, IL-10, IL-17, and VEGF are involved in this process, leading to the increased activity of superoxide dismutase (SOD). Oxidative stress has proven to be a hallmark of the disease [1,7,8,9].

The purpose of this manuscript is to establish the importance of pro- and anti-inflammatory molecules in endometriosis, based on the current literature, with the main objectives of identifying useful biomarkers for the detection of endometriosis and exploring a new therapeutic approach for curative purposes rather than merely symptom relief. We can affirm that pro-inflammatory and anti-inflammatory cytokines play roles in survival, growth, differentiation, invasion, angiogenesis, and immune escape, which provides a perspective for approaching future clinical implications and therapy [7]. They are also studied in the desire to identify possible non-invasive serum markers. However, as for treatment, we have insufficient control over anti-inflammatory drugs, which come with multiple adverse effects and inhibit the immune response, resulting in an increased risk of systemic infections [2]. 

## 2. Materials and Methods

The objective of our study is to evaluate cytokines as biomarkers for triage tests to assess their diagnostic utility and their utility as targeted treatments, knowing that there is no cure for endometriosis.

We assessed published peer-reviewed studies, in vitro and also in vivo, on animals and humans using the key words (endometriosis; ectopic endometrial tissue; pro-inflammatory cytokines; pathogenic mechanism; anti-inflammatory cytokines), including randomized controlled trials, observational studies, or previous review studies in the field of the topic. The period of the database that has been searched is from 2014 to 2023.

We excluded case reports or case series, studies reported only in abstract forms, studies based on observational laparoscopy without histological confirmation, and studies published in other languages than English (Figure 1).

Then we structured the manuscript into proinflammatory and anti-inflammatory cytokines. A short chapter is dedicated to future treatment; this is another major objective of the studies, as there is no specific therapy yet for endometriosis. Some cytokines show their value in this sense if they have no value as biomarkers. The main etiopathogenic mechanism is menstrual reflux. Here, we included a brief reference to the anatomy and immunological cells present in the endometrium that are possibly involved in endometriosis.

Being a narrative review, it has some limitations; for example, the methodology is not as restrictive as in a systematic review, and it is less reproducible, which can potentially lead to bias.

## 3. Anatomy and Immunology

The endometrium is structured into a basal layer and a functional layer (Figure 2). Throughout the stroma and between the epithelial cells of the functional layer, immune cells are scattered. The composition of these immune cells is heavily influenced by hormonal changes. Lymphoid aggregates form in the basal layer during the proliferative phase, consisting of B cells and cytotoxic CD8+ T cells, and they are surrounded by myeloid cells. T cells make up the majority of the leukocytes, followed by NK cells and macrophages [10].

During the secretory phase, there is a decline in T cell levels, with the majority now comprising NK cells and macrophages. Neutrophils exhibit high levels of VEGF expression and contribute to an inflammatory environment [10].

Immunological changes occur in endometriosis. Mast cells are the key players in allergic responses, but they are also implicated in angiogenesis, fibrosis, and pain in endometriosis. The influence of estrogen on mast cell function is a potential factor in the pathophysiology of allergic and chronic inflammatory diseases. The McCallion et al. (2022) study shows that endometriotic lesions had significantly higher levels of stem cell factors, growth factors, and mast cell expression [11]. Also, an aberrant transcriptome of fallopian tube epithelium and microenvironment changes causes of cytokines in tubal fluid are possible cause for tubal endometriosis [12]. In the Hang Qi et al. (2020) study, 15 pathways were discovered that induce differential regulation of cytokine production in macrophages and T-helper cells by IL-17A and IL-17F. Also, hypoxia induced the upregulation of IL-6 and TNF-alpha [12]. High activity of plasma cells was discovered, and a modified ratio of Th1/Th2 with increasing Th2 increased the number of mast cells because of the significantly higher level of stem cell factor [11,12].

## 4. Pro-Inflammatory Cytokines

Cytokines, both proinflammatory and anti-inflammatory, were discovered in endometriosis biopsy specimens, and it was already postulated that they are involved in the etiopathogenesis of this disease, which is more studied in cancers. A study evaluated, alongside IL-1, another pro-inflammatory cytokine observed in endometriosis, the macrophage migration inhibitory factor (MIF). MIF plays a regulatory role in the immune response, angiogenesis, and excessive estrogen production [13].

Cytokines have various mechanisms of action with various effects in endometriosis, related to their involvement in pain, embryonic implantation, and angiogenesis, all related to oxidative stress and implicated in IL-8 and IL-12, but they need to be validated for significance, specificity, and sensitivity [14].

A non-invasive diagnostic test for endometriosis using the antibody array approach was conducted, and IL-31 showed potential as a possible biomarker for endometriosis. The lack of non-invasive diagnostic tests contributes to a diagnostic delay of 8–11 years. In order to reduce this delay, a non-invasive diagnostic using biomarkers is needed. Until now, the most frequent marker was CA-125, but there is a lack of sensibility and specificity [15]. This is because various proinflammatory cytokines, such as IL-17 and IL-33, are also found in both endometriosis and cardio-vascular diseases [16]. 

Studies have investigated the relationship between endometriosis and ovarian cancer. Cytokines such as IL-2, IL-5, IL-6, IL-8, and IL-10, both pro-inflammatory and anti-inflammatory, have been measured in serum, intracystic fluid, and peritoneal fluid in endometriomas and ovarian cancers. The aim has been to determine the optimal cut-off point for serum cytokines to differentiate between patterns of ovarian malignancy and endometriosis. It has been demonstrated that significantly elevated levels of IL-6, IL-8, and IL-10 are exhibited in cancers. But the cut-off values for these cytokines in serum were 5.3 pg/mL (IL-6), 56.2 pg/mL (IL-8), and 12.56 pg/mL (IL-10) [5].

Oxidative stress has been suggested as a potential factor associated with the progression of the disease. One study assessed the diagnostic performance of high mobility group box-1 (HMGB1) and toll-like receptor 4 (TLR4), which seem to be associated with this process of damage-associated molecular patterns (DAMPs) and induce endometriosis. Immunohistochemistry has revealed their presence in biopsy specimens, both in epithelial and stromal cells. It appears that NF-kB inhibitors and TLR4 agonists lead to the suppression of HMGB1 and decreases in IL-6 levels, demonstrating their involvement in endometriosis and the clinical relevance of this new finding. Additionally, HMGB1 plays a physiological role in the nucleus, but when secreted extracellularly, it acts as a damage-associated molecular pattern, triggering an inflammatory response and progesterone resistance [17]. It is associated with increased levels of IL-6, TNF-alpha, and IL-1 beta, representing another pathogenic mechanism associated with the development of endometriosis. It shows potential clinical applications by targeting HMGB1 in endometriotic cells. HMGB1 levels increase under hypoxic conditions, thus involving ROS [18].

A study evaluated the accuracy of immunohistochemical and immunofluorescence analyses, which showed the presence of interleukin-1 receptor type 1 (IL-1RI) in endometriotic tissue, especially in the glands, and also in endothelial cells, macrophages, and T-cells in typical black-blue endometriotic tissue. It is also found in red endometriotic implants, which are highly vascularized, showing a relationship with the activity of the disease and an involvement in endometriotic tissue growth, development, and oxidative modifications [1]. 

Cytokines are found in intrafollicular fluid (Table 1). They have an impact on telomeres and mRNA expression in endometriosis. In patients with endometriosis, there is a significantly reduced number of antral follicles and a decreased number of oocytes retrieved through punctures. Among these retrieved oocytes, only a few have been mature and of optimal quality. It appears that increased levels of NF-kB and TNF-alpha in follicular fluid have a negative influence on the quantity and quality of oocytes [19]. Some studies have evaluated the follicular fluids of patients with endometriosis undergoing IVF, assessing the levels of IL-5, IL-6, and IL-3. It has been observed that there is a failure in the immunological defense system in endometriosis, but these cytokines do not have relevance as biomarkers [20].

Regarding infertility, the role of vitamin D has been studied, including its concentration in follicular fluid. An inverse proportion between vitamin D and IL-6 has been found. Additionally, there appears to be a correlation between vitamin D and other inflammatory factors such as TNF-alpha, IL-1 beta, IL-6, IL-8, and IL-10, as well as its involvement in autoimmune diseases and cancers. However, the association with endometriosis and clinical pregnancy rates has yielded inconclusive results [21]. Nevertheless, it clearly influences maternal–fetal communication and fetal development without being able to demonstrate their relevance as biomarkers [22].

Macrophages remain the most prominent immune cells observed in endometriotic cysts, as they are responsible for the production of IL-6 and TNF-alpha, according to another study [23]. These cytokines function as factors involved in the carcinogenesis of ectopic endometria, particularly clear cell carcinoma [23].

IL-6, IL-10, and TNF-alpha are implicated in the growth of endometriotic stromal cells. An additional study specifically addressed ovarian endometriosis and the involvement of these cytokines, and we present its findings [23].

They are pro-inflammatory cytokines and are part of the inflammatory status associated with the carcinogenesis of endometriosis. Local hypoxia, the production of reactive oxygen species, and iron overload are involved. The chocolate fluid in endometriotic cysts contains blood and cytokines that are implicated in both the carcinogenesis process and infertility. Macrophages infiltrate the cyst wall and are the main component of inflammatory cells. Infiltration is more pronounced during episodes of fresh hemorrhage, as only fresh hemoglobin stimulates increases in IL-6, which is essential in cancer [23].

The stromal cells of an endometriotic cyst, upon exposure to the cyst’s fluid, undergo ferroptosis, which, surprisingly, triggers the release of angiogenic growth cytokines, such as VEGF-A and IL-8, in endometriotic cells. Iron overload, along with genetic and epigenetic factors, is implicated in this process. Small, dysmorphic mitochondria are closely associated with iron accumulation. Ferroptosis is modulated by the intracellular iron overload resulting from repeated episodes of bleeding [24]. These findings could be the basis for future targeted treatments.

Another mechanism implicated in infertility is the compromise of embryo implantation, a well-known factor in pelvic inflammatory disease, polycystic ovary syndrome, and endometriosis [25,26]. Dysbiosis of the endometrial microbiota and pro-inflammatory cytokines such as IL-6, IL-8, and IL-17 are implicated in infertility and in cancer (Table 1). And a study evaluated the accuracy of these findings [25].

The effect of cytokines, angiogenesis, and extracellular matrix degradation augmented by oxidative stress on the pathogenesis of endometriosis remains unclear. Amalesh Nanda et al. (2020) demonstrated in his study that VEGF, MMP2, MMP9, and cyclooxygenase COX 2 were higher in endometriosis, but IL-10 was the most significant variable capable of discriminating endometriosis samples from controls, being considered a potential biomarker [27].

According to another study, extracellular vesicle (EV)-associated VEGF-C secreted by proinflammatory cytokine-stimulated endometriotic stromal cells is a critical modulator of endometriosis by promoting lymphangiogenesis. Invaded lymphatic vessels may serve as a canal for the infiltration of immune cells, which enhances the inflammatory status of endometriosis. VEGF-C can be a non-invasive diagnostic biomarker and a potential therapeutic target for endometriosis. VEGF-C is upregulated by IL-1 beta, and TNF-alpha extracellular vesicles may serve as cargos that carry and deliver VEGF molecules to the remote lymphatic vessels. In this study, we used a nanoparticle tracking assay [28].

There is a genome-wide association study of endometriosis that identified independent loci SNPs (single nucleotide polymorphism) at the IL-1A gene locus, which is associated with endometriosis risk. IL-1A is implicated in the pathogenesis of endometriosis [29].

Common genetic alterations, such as PTEN, p53, bcl gene mutations, and ARID1A mutations, have been established for both endometriosis and cancers. A chronic inflammatory state leads to cytokine release, followed by the unregulated mitotic division, growth, differentiation, migration, and apoptosis, similar to malignant mechanisms [6]. Tripartit-motif-containing 24 (TRIM24) appears in inflammation associated with cancers, while the NACHT, LRR, and PYD domain-containing protein 3 (NLRP3) inflammasome are implicated in endometriosis. The relationship between them is mediated by IL-1 beta. TRIM24 is inversely proportional to the progression of endometriosis [30].

Repeated tissue injury, repair, and fibrosis play a pivotal role in endometriosis. Fibrotic tissue consists of extracellular matrix proteins, regulated by transcriptional factors. Periostin is a key extracellular matrix protein. Periostin and transcription factor 21 TCF21 is not detected in the stromal cells of women without endometriosis, but it is strongly detected in deep endometriosis. One study evaluated how IL-4 and IL-13 increase the expression of periostin and TCF21, which are involved in the regulation of fibrosis in endometriosis. TCF21 may be a promising therapeutic target and biomarker in endometriosis [31].

On the other hand, there are studies like that led by Tamara Knific (2019) that have shown no differences in the concentration of the measured 40 cytokines between patients and controls, demonstrating that the panel of cytokines used in their study were not relevant biomarkers [32].

## 5. Anti-Inflammatory Cytokines

Furthermore, the reciprocal influence between anti-inflammatory and pro-inflammatory cytokines and their role in the pathogenesis of endometriosis has been assessed. One study showed that IL-37, an anti-inflammatory factor primarily produced by T-helper cells, acts as a trigger for pro-inflammatory factors such as IL-6, IL-8, and VEGF, thereby participating in the pathogenesis of endometriosis and enhancing angiogenesis. Further studies are needed to provide evidence for this mechanism [2]. IL-37 is produced by numerous cells, including stromal cells, fibroblasts, and endothelial cells. The heterogeneity of endometriotic lesions makes it challenging to identify the specific cells involved in this mechanism. However, it has been observed that the level of IL-37 decreases by simply removing endometriotic lesions, so it can probably be used as a biomarker [2].

A study revealed there are potential differences in the immune profiles between women with and without endometriosis. IL-13 is lower in the endometriosis group, according to the study by H. Jorgensen et al. (2017), but future analyses of the pathophysiological mechanisms of endometriosis, including dysregulated Th1/Th2 responses, are needed [33].

Anti-inflammatory cytokines produced by T-helper 2 cells, such as IL-4, IL-10, and IL-13, have been investigated in endometriotic lesions to understand their roles. Other innate cytokines produced by thymic stromal lymphocytes, such as IL-25 and IL-37, have been found in peritoneal fluid in cases of endometriotic lesions (Table 1). It has been observed that while pro-inflammatory cytokines increase during the progression of the disease, anti-inflammatory cytokines increase in the advanced stages. Researchers are trying to determine their value in the diagnosis of the pathology. These cytokines originate from immune cells, endometrial epithelial cells, endometriotic mesenchymal stem cells, peritoneal mesothelial cells, and platelets. Their established roles include the regulation of immunity, inflammation, cell proliferation, apoptosis, adhesion, and migration, and they have crucial roles in epithelial–mesenchymal transition (EMT), fibroblast-to-myofibroblast differentiation, smooth muscle metaplasia, fibrogenesis, and angiogenesis. All these phenomena have been observed in biopsies and cell cultures [2].

One study revealed that prostaglandin E2 is also involved in endometriosis, stimulating P450 aromatase and increasing estrogen production in endometriotic tissue, as well as enhancing the Th2 immune response. It also increases local estrogen production [2].

IL-37 is an anti-inflammatory cytokine found in the serum and peritoneal fluid of patients with endometriosis. At the same time, TNF-alpha in the peritoneal fluid has been evaluated, and it increases and shifts helper T cells towards Th2, leading to increased levels of IL-37 and IL-33. NF-kB is also activated, and ICAM-1 expression in endometriosis confirms the inflammatory pattern, according to Kaabachi’s study (2017). IL-33 and IL-37 belong to the IL-1 family, though with an unclear role [34]. IL-37 is a relatively recently discovered cytokine, with its most biologically active isoform being IL-37b, and it has been studied in cancers where it appears to have a protective role. However, it has also been found in endometriosis and adenomyosis, with a suppressor role in the control of cell proliferation, inhibiting migration, and invasion and decreasing the expression of matrix metalloproteinase 2 (MMP2) via the Rac1/NF-kB signaling pathway. It does not, however, influence the epithelial–mesenchymal transition. It attenuates the occurrence of tumor metastases and can be used as a novel target for the diagnosis and treatment of malignant transformations in endometriosis [35].

Disturbed T-cell interactions have been observed in endometriosis. The defective expression of ICAM-1 (CD54) on secretory endometrial cells indicates the interaction between pro-inflammatory and anti-inflammatory mediators in the development of endometriosis [34].

IL-4 inhibits the expression of IL-37. IL-37 has been studied in chronic autoimmune diseases and cancers. It appears to affect gene transcription through its binding to smad 3. Recently, it has also been investigated in endometriosis biopsy specimens [34].

IL-1 receptor accessory protein (IL-1RAcP) plays a role in signaling IL-1 family cytokines, including IL-1, IL-33, and IL-36, which are implicated in chronic inflammatory diseases, autoimmune diseases, endometriosis, type 1 diabetes, and preeclampsia. It can be proposed as a treatment by utilizing the inhibitory properties of IL-1RAcP in inflammation [36]. The pro-inflammatory and anti-inflammatory cytokines are presented in the table below (Table 1).

**Table 1 diagnostics-14-00312-t001:** The roles of anti-inflammatory and pro-inflammatory cytokines, their efficacy, mechanism of action, clinical site of action, and mode of action.

	Cytokine Type	Efficacy	Mechanism	Clinical Site	Mode of Action
Anti-inflammatory cytokines	IL-4 [2] IL-13 [33] IL-25 [2] IL-37 [2,34,35] IL-33 [16,34] IL-36 [36]	- Treatments - Interaction between pro-inflammatory and anti-inflammatory mechanisms in endometriosis - Affects gene transcription	- Gene mutation implicated - ROS - Increased estrogen production - Progesterone resistant	- Peritoneal fluid - Endometrial cells - Endometrial tissue - Platelets - Biopsy specimens	- Regulation of immunity - Inflammation - Cell proliferation - Apoptosis - Adhesion - Migration - EMT - Fibroblast-to-myofibroblast differentiation - Smooth muscle metaplasia - Fibrogenesis - Angiogenesis - Decrease MMP - Reduces the occurrence of cancer metastases
Pro-inflammatory cytokines	PGE2 [2] IL-1 [1,28,29] MIF [13] IL-6 [2,14,18,21,23] TNF-alpha [18,19,21,23] IL-8 [14,25] NF-kB [17,19] IL-5 [5,20] IL-3 [20] IL-17 [25] IL -10 [5,21]	- No relevance as biomarkers - Beneficial for discovering treatments	- Involving ROS - Iron overload - Gene mutation implicated	- Modifications in biopsy specimens of endometrial tissue - Found in intrafollicular fluid - Alters the microbiota of ectopic endometria and uterine cavities	- Modifies telomeres and mRNA in endometriosis - Failure in the immunological defense system in endometriosis - Infertility with influences on maternal–fetal communication - Carcinogenesis - Embryo implantation - Apoptosis, migration, adherence, proliferation, and embryo implantation - Angiogenesis, adherence, and proliferation

## 6. Malignant Transformations in Endometriosis

Malignant transformations in endometriosis are rare, occurring in only 1% of cases. However, endometriosis is four times more likely to develop malignancies compared to 20 years ago and compared to the occurrence of de novo ovarian cancer in patients without endometriosis. CTs and MRIs are useful in combination with histopathology for diagnosis [37].

There is a correlation between inflammation, endometriosis, adenomyosis, cardiovascular diseases, and systemic complications that extend beyond local effects. This involves genetics, epigenetics, sex steroids, extracellular matrixes, stem cells, cardiometabolic risk factors, diet, vitamin D, and the immune system. All of these factors contribute to impaired fertility, potential malignancies, or systemic diseases. Inflammatory markers and imaging molecules are increasingly being studied as non-invasive techniques for prompt detection and future treatment [13]. All this effort by the researchers in this direction is due to the lack of non-invasive diagnosis, which postpones obtaining the correct diagnosis for even ten years. In addition, the conventional treatment resulted in the amelioration of the disease but without a curative effect.

It is generally established that endometriosis is an estrogen-dependent disease. An increase in estrogen synthesis and progesterone resistance, which leads to hormonal imbalance, is associated with endometriosis. Treatment of this pathology with progestins induces a rapid increase in IL-13 and IL-15 expression. But long-term treatment with progestins does not have a sustained effect [2]. The estrogen-rich environment created by endometriotic lesions is also shown to regulate the production of IL-6, promoting cell proliferation [6]. It is also known that they have an important role in malignant transformation [20].

Endometriosis shares a common inflammatory pathogenesis with cancers, involving TNF-alpha, IL-1, IL-6, IL-8, and VEGF, associated with retrograde menstruation, iron overload, and the activation of macrophages, triggering aberrant inflammatory signaling and impairing phagocytic potential. The local inflammatory environment leads to disease progression and stimulates angiogenesis, with systemic effects. The overexpression of cyclooxygenase 2 results in the secretion of a large amount of PGE2, which activates NF-kB. PGE2 is a mediator of endometriosis that maintains the proliferation and viability of endometriotic cells. Alongside PGE2, IL-1, and VEGF, macrophage migration inhibitory factor (MIF), a pro-inflammatory factor with a regulatory role in the immune system, angiogenesis, proliferation, and estradiol synthesis, also plays a role. In conjunction with iron overload, it activates peritoneal macrophages, stimulating proliferation, adhesion, and angiogenesis and contributing to disease progression [13].

Progesterone resistance has been hypothesized to further contribute to a pro-inflammatory phenotype in the context of endometriosis. Endometriotic lesions exhibit limited progesterone receptor expression due to gene polymorphism, altered miRNA expression, and epigenetic modifications to the progesterone receptor. Thus, the increased activation of NF-kB and decreased progesterone receptor immunoreactivity can serve as histological biomarkers for recurrent ovarian endometriosis with potential malignancy [13,37].

Numerous pathways and regulators are shared by tumors and endometriosis. Their alteration may promote the progression of malignancies as well as the development of gestation. B-cell-lymphoma 6 (BCL6) is a key oncogene, serving as a master regulator of humoral immunity and lymphoma survival, and it also plays an important role in trophoblastic cell function. It is upregulated in preeclampsia placentas as well as in endometriotic lesions, inhibiting cell terminal differentiation. Various mechanisms are involved, including altered gene transcription, deregulated epigenetics, gene expression in response to stress, heat shock proteins 90 and 1, and posttranslational modifications such as phosphorylation or methylation. Inflammation is also a contributing factor. BCL6 has been discovered in endometrial mesenchymal stem cells and bone marrow mesenchymal stem cells, conferring migration and angiogenesis abilities. In the endometrium, it aids in self-renewal during the menstrual cycle, while in endometriosis, it is associated with increased proliferation, enhanced inflammation, decreased apoptosis, deregulated immune responses, and progesterone resistance [38].

Chronic inflammation may be linked to the tumorigenesis associated with endometriosis. IL-6, IL-8, IL-10, and TNF-alpha induce the polarization of dendritic cells, NK cells, and monocytes, as well as alterations in the inflammasome and complement system. However, the exact immunological mechanisms, pathways, and cellular processes involved remain unknown [10].

A differential diagnosis can be attempted between benign serous ovarian cysts, endometriomas, and ovarian cancers using the levels of IL-10 and TNF-alpha. Serum levels are significantly elevated in cancers, moderately increased in cases of endometriosis, and decreased in benign cysts, according to Robinson’s study (2020) [37]. Therefore, as malignancy progresses, the values increase. Furthermore, in carcinomatous ascites, the peritoneal levels of MMP, VEGF-A, TNF-alpha, and IL-2 increase. Human leukocyte antigen (HLA) is a major histocompatibility complex (MHC) antigen expressed on a cell’s surface. Its expression is influenced by environmental factors such as hypoxia, stress, hormones, cytokines, and viruses. This results in the inactivation of cytotoxic T cells and a lesser production of antitumor antibodies. IL-10 is synthesized by activated T cells (Treg) in response to interactions with HLA-G. IL-10 serves as an important immunosuppressant, and its elevated concentrations lead to immune escape in certain malignancies. However, there are studies that have shown that the serum levels of IL-10, TNF-alpha, and soluble HLA-G (sHLA-G) alone are not sufficient for differentiating between endometriosis and cancer, despite being significantly increased compared to their levels in benign lesions [35,39].

Multiple associations of endometriosis with other pathologies involving inflammatory factors have been discovered. People with endometriosis experience an increased hypercoagulability status caused by inflammation, which poses a risk for atherosclerosis and vascular complications [4]. Additionally, there is an association with cancers, chronic inflammatory diseases, autoimmune diseases (ulcerative colitis, lupus erythematosus, rheumatoid arthritis, celiac disease, multiple sclerosis, etc.), and endocrine disorders [36]. Fibrinogen could be involved as a marker of inflammation and thrombosis, along with IL-17 and IL-33, which influence the innate and adaptive immune systems. They also play a role in tumorigenesis, involving ROS and NOS, which are synthesized in endothelial cells and lead to altered vascular permeability. Dyslipidemia may be implicated, and platelet aggregation has been reported [4,36].

In some studies, the prognostic value of the expression of certain inflammatory factors, such as IL-6, has been evaluated for clear cell carcinoma [40,41].

The risk of endometrial cancer associated with endometriosis has not been determined. A population-based observational study conducted by Hoon Kim (2022) established that the risk of endometrial cancer was found to be higher in women with endometriosis. The presence of endometriosis did not affect overall survival in those women [42]. It is important to establish the role of immune cells and the tumor microenvironment, in which inflammatory cytokines have a significant role, in the development and progression of the malignant transformation of endometriosis in order to explore innovative approaches for identifying effective drugs [43].

## 7. Future Treatment

Treatment of endometriosis is widely presented in the ESHRE Guidelines for endometriosis. It is subdivided into the treatment of pain, infertility, and surgical treatment. Here, we only do a brief recapitulation, as the purpose of the manuscript is to find other new types of targeted treatment. We mention here analgetics, combined hormonal treatment and contraceptives, progesterone, GnRh agonists, GnRh antagonists, and aromatase inhibitors). Infertility treatment consists of ovarian suppression and hormone therapies associated with surgery. Also, the most important treatment is surgery: ablation, excision of endometriotic lesions, surgical interruption of pelvic nerve pathways, surgery for deep endometriosis, and hysterectomy for endometriosis-associated pain [41].

Researchers are focused on new treatments that have a curative effect, while the ones we already know only alleviate the disease. Hence their focus on these treatments, starting with the pro-inflammatory and anti-inflammatory cytokines that have given results in cancer, the two diseases having similar behavior. The studies are under investigation.

There are no management guidelines for the prevention or treatment of endometriosis-related ovarian cancer [23]. Hence, there is intense concern about discovering a coherent treatment approach. In this regard, the role of nonsteroidal anti-inflammatory drugs (NSAIDs) and antibiotics, such as clarithromycin, has been identified to increase in IL-10 in endometriotic lesions, which plays a role in inhibiting cell proliferation. However, further studies are needed. CTLA-4 antibody, used in mice, and Tanshinone, an active diterpenoid extracted from plants used in traditional Chinese medicine, have been found to inhibit platelet aggregation and suppress the TGF-beta and NF-kB signaling pathways, thereby reducing fibrogenesis in deep endometriosis and suggesting potential therapeutic utility. Ginseng also reduces fibrosis in endometriosis and decreases Ki-67, fibronectin, MMP2, and MMP9. Epigallocatechin-3-gallate (EGCG), a polyphenol found in green tea, reduces endometriosis, implantation, and angiogenesis [2]. However, anti-inflammatory treatment has proven to be ineffective thus far. Individualized therapy is key [2].

Li G. and co., in their study (2022), showed that deferoxamine is an iron chelator that could be injected into endometriotic lesions to reduce inflammation and proliferation [24]. It is an inhibitor of ferroptosis, which is found in endometriosis biopsy specimens along with VEGF and IL-8, as well as in cell cultures [24].

Another study observed that low doses of (the sequential kinetic activation signaling molecule) SKA-progesterone in combination with SKA-IL-10 inhibit NF-kB and have a beneficial effect on endometriosis (SKA refers to sequential kinetic activator) [44]. These are molecules administered orally over a long period of time. Hormones and cytokines in ultra-low doses act as “signaling molecules” capable of exerting immune–endocrine modulation. A new approach has been proposed: the use of low doses of signaling molecules such as cytokines, neurotransmitters, and hormones. This new pharmaceutical technology is called sequential kinetic activation, which is a drug delivery system that utilizes the minimum effective dose to achieve therapeutic results. This approach represents a low-dose medicine that modulates intracellular transcription mechanisms due to the low concentrations of other messenger molecules [44].

Lyu D, in his study (2019), observed that INT 777, a TGR5 agonist, has a protective effect against inflammation and oxidative stress in endometriotic stromal cells [45]. Endometrial stem cells (ESCs) are capable of regenerating the endometrium within a month. ESCs mediate angiogenesis and stromal regeneration, activated by TGR5. This represents a new therapeutic approach based on regenerative medicine and cell-based therapy [45].

The poor response to conventional platinum-based chemotherapy and the poor prognosis of clear cell carcinoma developed from endometriotic cysts have led to intense studies in search of alternative adjuvant therapies. In early-stage tumors, adjuvant therapy may not be required. Genetics is implicated in this context, with ARID1A remodeling being a key driver for clear cell carcinoma. It is known as a pathogenic mechanism in endometriosis and chronic inflammation. IL-6 is also implicated and may serve as a potential therapeutic target [46].

Vorinostat is another experimental treatment used to prevent the malignant transformation of endometriotic lesions. It decreases M2 macrophage polarization through ARID1A. ARID1A mutations play an important role in endometriosis by mediating the expression of HDAC6, although the exact pathogenic mechanism is unclear. The ARID1A/HDAC6-induced M2 polarization of macrophages is mediated through IL-10. The HDAC inhibitor Vorinostat plays a role in inhibiting cell growth. ARID1A (AT-rich interactive domain 1A) is a factor that regulates chromatin remodeling. It is considered a cancer suppressor gene that prevents cells from growing and dividing quickly or uncontrollably. The ARID1A mutation is associated with the progression of certain cancers [47].

Transcription nuclear factor kappa B is a gene whose expression is dependent on estrogen, but it is differentially regulated in ectopic endometria compared to normal endometria. Ectopic endometria exhibits reduced immunological reactivity to kappa B inhibitors compared to normal endometria [48].

In addition to established treatments aimed at symptom relief rather than cures, efforts are being made to explore the use of immunomodulators, cardiometabolic medications, cyclooxygenase-2 (COX-2) inhibitors, and NF-kB inhibitors such as celecoxib, rofecoxib, dexketoprofen, and tromedadol, which can target the inflammatory pathways involved in the pathogenesis of endometriosis [13].

As novel therapeutic targets for endometriosis, oral administration of an inhibitor against IL-1 receptor-associated kinase 4 IRAK4 also suppresses lesion formation [49].

Cardiometabolic drugs such as metformin and thiazolidinediones have been used to reduce the size of endometriotic implants. Statins, commonly prescribed as antihyperlipidemic medications, have also been studied for their anti-inflammatory effects on endometriosis. Additionally, anti-angiotensin II blockers have shown anti-inflammatory properties [13] (Table 2).

## 8. Discussion

Endometriosis is a chronic inflammatory disease with a prevalence of 5–10% in women of reproductive age that accounts for the pain experienced by sufferers, and oxidative stress plays a significant role in its pathogenic mechanism [41]. Ongoing research is centered on identifying novel mechanisms and pathways that connect immunological changes to ectopic endometrial tissues and the involvement of cytokines [2].

This inflammatory component of endometriosis associates it with other inflammatory pathologies, finding common ground in biomarkers and treatment strategies.

Many biomarkers have been studied first in cancer and then in endometriosis. Some cytokines show increased levels in both pathologies [5]. They are studied in order to identify non-invasive biomarkers.

As we have shown, cytokines have various mechanisms of action with involvement in embryonic implantation, pain, and fibrosis, and depending on what symptoms the patient presents, we focus on relevant biomarkers (Table 1). New biomarker panels for disease detection are needed due to their non-specific symptoms and lack of non-invasive diagnosis. As we can see in Table 1, we can identify biomarkers, for example, IL-10, whose mechanism of action focuses on angiogenesis and adhesion, which differentiate endometriosis from other pathologies [27].

In other cases, proteomics whose expression increases under the influence of interleukins 3 and 4, such as TCF21, can be used either as a biomarker or as a targeted treatment [31]. The presence of IL-R1 in all stages of endometriosis, including the early stage, makes this receptor a possible biomarker or targeted treatment [1].

The anti-inflammatory cytokines, whose levels have been found to increase in the advanced stages of the disease, seem to have a protective role in endometriosis. They have a suppressive role in cell proliferation [2]. They can be proposed as a treatment due to their inhibitory effect.

Multiple associations of endometriosis with other pathologies involving inflammatory factors have been discovered. Among the cytokines, IL-33 is a member of the IL-1 family and has been studied in myocardial infarctions. Endometriosis and cardiovascular disease share mutual pathogenic mechanisms and genetic susceptibility [50]. These associations with other pathologies create the premise for the use of biomarkers or targeted treatments from other pathologies in the diagnosis and treatment of endometriosis.

According to Beste’s study (2014), there is a crucial involvement of the macrophage inflammatory network in elevated levels of cytokines that may improve risk stratification for better targeted treatment [51]. But according to other studies, hs-CRP and other studied inflammatory biomarkers have no value for the assessment and diagnosis of endometriosis [52].

However, genetic and epigenetic factors influence the inflammatory changes in endometriosis [53].

The lack of a non-invasive diagnosis delays the correct diagnosis for many years. Endometriosis shares a similar behavior with cancer, even in terms of pathogenesis. This is the reason why the researcher’s initiative to study cancer biomarkers along with targeted treatments for endometriosis seems reasonable. As described in Section 6, additional studies are still needed to reach the correct conclusion.

Regarding treatment, considering the fact that endometriosis is a challenge due to the wide range of clinical manifestations, current treatments primarily focus on providing symptomatic relief rather than targeting the underlying disease mechanism. We emphasize that there is no curative treatment. Common treatments aim to reduce inflammation and remove ectopic lesions, but recurrence and therapy resistance remain significant issues. The purpose of this review is to present new types of targeted treatments that could have a curative effect. As we presented in Section 7, the targeted treatments still need in-depth studies, and most of them have been studied and are used in other pathologies, especially cancer.

Complex events occur in endometriosis. Unlike normal endometria, endometriotic stromal cells express abnormally high levels of P450 aromatase, induced by PGE2, and this leads to increased estrogen production. This excessive production of estradiol is now recognized as one of the key factors in the pathology of endometriosis, where estrogen and inflammation interact in a complex manner. The disease involves 26 inflammation-related genes, and TNF-alpha, IL-1, MMP-1, MMP-2, MMP-3, MMP-9, and NF-kB alter the extracellular matrix. Stem cells are recruited to the peritoneal cavity, contributing to processes such as angiogenesis, adhesion, and migration [13].

The immune mechanism is also involved, with Treg cells and intercellular adhesion molecule 1 (ICAM-1) playing roles. Treg cells secrete inhibitory cytokines that suppress the immune response by NK cells, macrophages, and CD4+ and CD8+ lymphocytes [13]. All these are still being researched in order to find useful biomarkers and applicable treatments.

The aspects related to repeated attempts to detect pro-inflammatory and anti-inflammatory molecules that are useful as biomarkers in the non-invasive detection of endometriosis have been presented in previous chapters, and none have obtained satisfactory results, according to the data in the literature. However, there has been progress in terms of therapeutic innovations based on these discoveries of cytokine involvement in endometriosis. An important role has also been played by identifying common features with other common inflammatory diseases and cancers. However, further investigations are absolutely necessary, and the research is focused not only on the symptomatic treatment of pain (since endometriosis is a disabling disease) but also on curative purposes.

## 9. Conclusions

This narrative review of the literature explores the possibility of the involvement of pro- and anti-inflammatory molecules in the process of the appearance of endometriosis. Cytokines play an important role in the progression of the disease. Today, we can affirm that pro-inflammatory and anti-inflammatory cytokines are studied as biomarkers, with the results being promising, but more studies are needed.

Also, their role in endometriosis provides a perspective for approaching future clinical implications and therapy.

## Figures and Tables

**Figure 1 diagnostics-14-00312-f001:**
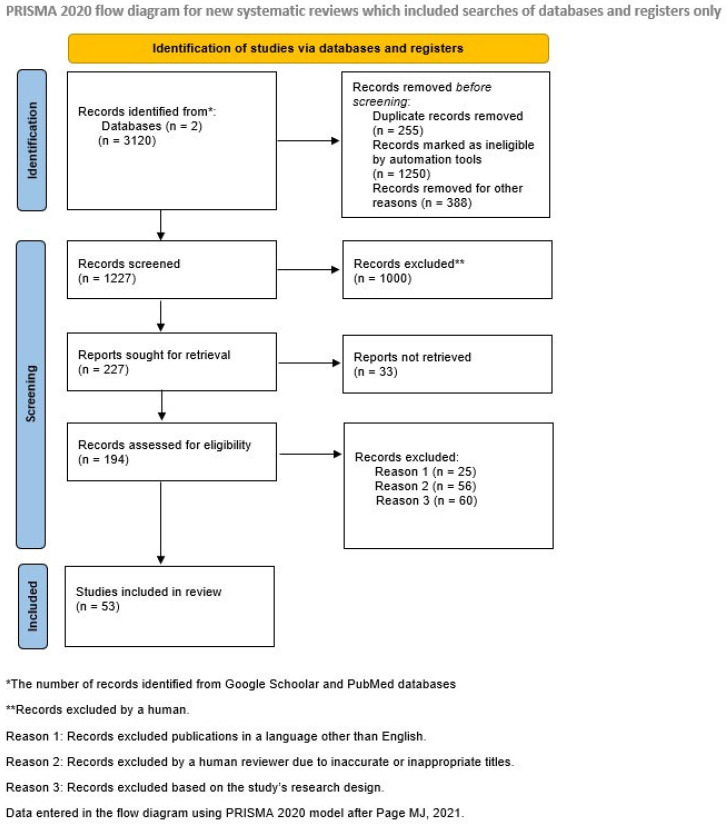
PRISMA flow diagram.

**Figure 2 diagnostics-14-00312-f002:**
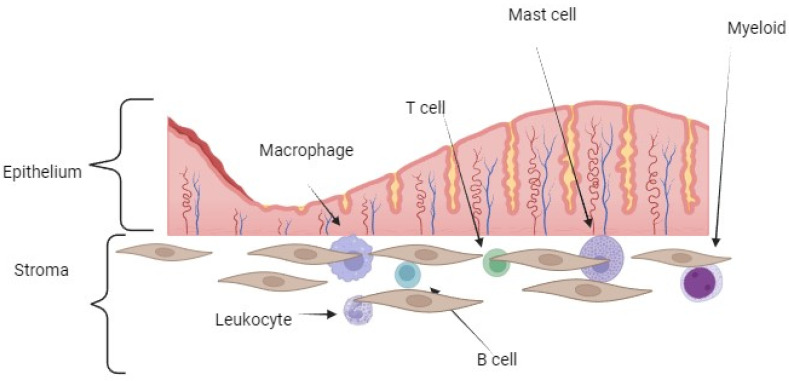
Schematic representation of endometrial immune cells who suffer changes in endometriosis and contribute to an inflammatory environment—infiltration of macrophages, myeloid, and mast cells during bleeding secondary release pro-inflammatory cytokines, especially mast cells. Macrophages produce IL-6, IL-4, IL-10, IL-2, and TNF-alfa. Mast cells are involved in humoral immunity. IL-37 and IL-17 are primarily produced by T-helper cells. Neutrophils and leukocytes contribute to an inflammatory environment, also producing cytokines. A modified ratio of Th1/Th2 with increasing Th2 was found, which secretes TNF-alfa.

**Table 2 diagnostics-14-00312-t002:** Clinical trials.

Treatment Attempts	Activity Mechanisms
Nonsteroidal anti-inflammatory drugs (NSAIDs), antibiotics such as clarithromycin [2]	Increasing IL-10 with antiproliferative effect
CTLA-4 antibody and Tanshinone [2]	Suppress the TGF-beta and NF-kb signaling pathways, thereby reducing fibrogenesis in deep endometriosis
Deferoxamine [24]	Reduces fibrosis, anti-inflammatory effect
SKA-progesterone in combination with SKA-IL-10 [44]	Inhibit NF-kb
INT 777 [45]	Protective effect against inflammation and oxidative stress in endometriotic stromal cells
Vorinostat [47]	Experimental treatment used to prevent the malignant transformation of endometriotic lesions
Immunomodulators, cardiometabolic medications, cyclooxygenase-2 (COX-2) inhibitors, and NF-kb inhibitors such as celecoxib, rofecoxib, dexketoprofen, and tromedadol [13]	Target the inflammatory pathways involved in the pathogenesis of endometriosis
Inhibitor against IL-1 receptor associated kinase 4 IRAK4 [49]	Supressed endometriotic lesion formation
Metformin and thiazolidinediones [13]	Reduce the size of endometriotic implants
Statins and anti-angiotensin II blockers [13]	Anti-inflammatory effects on endometriosis

## Data Availability

Not applicable.

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
