# Peer review of "Endometriosis and the Role of Pro-Inflammatory and Anti-Inflammatory Cytokines in Pathophysiology: A Narrative Review of the Literature"

_diagnostics, 2024, doi:10.3390/diagnostics14030312_

Round 1
Reviewer 1 Report (Previous Reviewer 2)
Comments and Suggestions for Authors all the points have been addressed.
Author Response
Review 1
Thank you very much!
Reviewer 2 Report (Previous Reviewer 3)
Comments and Suggestions for Authors
Title: I suggest to add the type of the review.
Materials and Methods - lack of:
a) terms used in searching;
b) Medline from 1997 is named PubMed (you have got 1 database);
c) previous review studies in field of the topic (line 107-108: "We assessed published peer-reviewed studies using the key words shown above, including randomised controlled trials, observational studies, or systematic reviews."),
d) the period (date of onset and end) of database which has been serched in your review;
e) the inclusion criteria are incomplete: human only or animal studies or in-vitro and in-vivo both human and animal
f) registration number (PROSPERO; see Item 24 in: https://www.bmj.com/content/372/bmj.n160);
g) study risk of bias assessment.
Figure 1 - records screened (n=192) and records assesed for elgibility (n=51) both have 141 records excluded - how canyou subtract (25+56+60=141) from records assesed for elgibility (n=51) and have still 51 records included in review? (51-141=-90)
Figure 1 - "PRISMA flow diagram" - add reference, please.
Figure 2 is still insuficient.
Table 1 - add references, please.
Comments on the Quality of English LanguageLine 515: "activator).[42] These" > activator [42]. These
Author Response
Reviewer 2
Dear reviewer:
- I added the terms we used as key words for the search.
- We used PubMed and Google Scholar, so I changed it in the legend.
- I changed in line 107-108 as you suggested.
- The period of database which has been searched is from 2014 to 2023.And I added this information in the manuscript line109.
- We included in-vitro studies and also in-vivo on animals and humans. We included this information in material and methods.
- It is a narrative review and do not require a registration number.
- Being a narrative review it has some limitations, the methodology is not so restrictive as in systematic review, is less reproducible, which can potentially lead to bias. [line 160-161].
Figure 1: I changed it all because it was from another article… I realized it now. I added in the figure the reference.
Figure 2: This is about the most important immune cells involved in endometriosis and released by menstrual reflux, and is not related to hormonal phases.
Table 1: I added references.
I made the change [line 505]
Thank you very much. Your suggestions were very helpful!
Reviewer 3 Report (New Reviewer)
Comments and Suggestions for Authors
The title of the manuscript, " Endometriosis and the Role of Pro-inflammatory and An-ti-inflammatory Cytokines In Pathophysiology: a Narrative Review of the Literature," is currently a hot topic in the field. The manuscript is very well written and co-related. However, the addition of a few more subtopics would greatly improve the manuscript.
1. I strongly suggest briefly describing the hormonal contribution to this disease as a subtopic, as it is well known to play a role in inflammation. And if possible elucidate it in a pictorial format.
2. Include a list of currently innovative ongoing clinical trials for endometriosis.
3. Also briefly describe the association of endometriosis with the high risks of endometrial hyperplasia and endometrial cancer via inflammatory cytokines.
4. The following articles are worth citing
a. Kim H, Kim HJ, Ahn HS. Does endometriosis increase the risks of endometrial hyperplasia and endometrial cancer? Gynecol Oncol. 2023 Feb;169:147-153. doi: 10.1016/j.ygyno.2022.06.021. Epub 2022 Nov 7. PMID: 36357191.
b. Dey DK, Krause D, Rai R, Choudhary S, Dockery LE, Chandra V. The role and participation of immune cells in the endometrial tumor microenvironment. Pharmacol Ther. 2023 Nov;251:108526. doi: 10.1016/j.pharmthera.2023.108526. Epub 2023 Sep 9. PMID: 37690483.
Author Response
Reviewer 3
Dear reviewer
- I included a brief paragraph about how hormones are involved in the inflammatory changes in endometriosis and in its malignant transformation , line 405-411: It is generally established that endometriosis is an estrogen dependent disease. An increase in estrogen synthesis and progesterone resistance which leads to homonal imbalance is associated with endometriosis. Treatment of this pathology with progestins induces a rapid increasing of IL-13 and IL -15 expression. But long-term treatment with progestins does not have a sustained effect.[2] The estrogen-rich environment created by endometriotic lesions is also shown to regulate the production of IL-6 promoting cell proliferation.[6] It is also known they have an important role in malignant transformation.[20]
- I added a table 2 related to the the topic you suggested.
- We briefly described the association between endometriosis and endometrial cancer and we added the references you suggested: The risk of endometrial cancer associated with endometriosis has not been determined. A populational-based observational study conducted by Hoon Kim (2022) established that the risk of endometrial cancer found to be higher in women with endometriosis. The presence of endometriosis did not affect overall survival in those women [42]. It is important to discover the role of immune cells and the tumor microenvironment, in which inflammatory cytokines have an significant role, in the development, progression of the malignant transformation of endometriosis in order to explore innovative approaches for identifying effective drugs.[43]( line 479-487)
Thank you very much for your suggestions.
Round 2
Reviewer 2 Report (Previous Reviewer 3)
Comments and Suggestions for Authors
The content is impressive however, the Figure 2 is still poor quality - it should depict your idea/ mechanisms, there is no important proinflammatory factors nor the medicines against it.
Comments on the Quality of English LanguageLine 603 - add space in 'table1', please.
Line 640 - 'chapter 7' => 'section 7'
Author Response
Reviewer 2
Dear reviewer:
I added the role of immune cells in endometriosis in the legend of Figure 2.
Also I made the changes you suggested.
Thank you very much.

Reviewer 3 Report (New Reviewer)
Comments and Suggestions for Authors
The present version of manuscript has been widely modified based on the reviewer’s prospective and has been greatly improved. Threfore, I suggest its publication in the present format.
Author Response
Reviewer 3
Dear reviewer
Thank you very much!

This manuscript is a resubmission of an earlier submission. The following is a list of the peer review reports and author responses from that submission.
Round 1
Reviewer 1 Report
Comments and Suggestions for Authors
In this review, Ioan Emilian Oală et. al, aimed to put together the role of pro- and anti- inflammatory cytokines in endometriosis with the purpose of identifying biomarkers that can be useful for the detection of endometriosis and exploring new therapeutic approach.
However, this is a very poorly organized review that mostly cites previous reviews instead of citing recent original research papers. Besides, throughout the review authors mention one topic in one paragraph and then mention completely different topic in the next paragraph without making a logical connection between topics. Authors needs to be more organized and make connection between paragraphs in all sections.
Moreover, the Figure 1 that meant to represent endometrium, is an inaccurate schematic presentation of endometrium that definitely needs to be revised and corrected. What is the pink layer indicated as stroma above functionalis layer of endometrium? Same pink layer is schematized below basal layer of endometrium? Is it another stromal layer instead of myometrium.
Reference #1 is an review that cites 2 papers dated 1992 and 2010 about the prevalence of endometriosis. Authors should provide an updated recent study to refer endometriosis prevalence, instead of citing another review.
Another inaccurate information is detected in introduction section lines 94-103. Reference #9 is about IL-17A not IL-37. Why authors mention IL37 instead of IL-17A?
In introduction section, in the aim paragraph, lines104-110, why authors citing a mice study that is related with abortion, in an review that the subject is endometriosis ? How they connect this abortion related mice study to the endometriosis related future clinical implications and therapy?
Anatomy section is about immune population of endometrium? Why titled as anatomy? Does authors need an anatomy section at all?
Overall this review, is a review of several previously published reviews , that decreases the originality. Besides, since this review has inaccurate information and organized very poorly, this reviewer cannot recommend for publication.
Comments on the Quality of English LanguageModerate editing needed.
Author Response
Cover letter 1
Thank you very much and about the topic, I agree with you that they are different depending on the research and it is difficult to make a connection between them, but I will try to do that. I added 12 recent original articles relevant to the manuscript.
I changed the figure 1.
As I said I added 12 more original articles relevant to the manuscript.
Reference 9 is related to line 98, reference 2 is related to line 101-108, reference 9 was used here by mistake.
I used reference 7 about the abortion in mice because the involvement of IL 17 in abortion in mice has a similar mechanism of involvement of IL 17 in embryo implantation in endometriosis in humans.
I added “ immunology “ to the chapter title. Anatomy is useful because the main etiopathogenesis of endometriosis, which everyone agrees on, is menstrual reflux, the cells of the immune system are also involved but which then undergo changes.
Also we used MDPI English editing.
Thank you very much and about the topic, I agree with you that they are different depending on the research and it is difficult to make a connection between them, but I will try to do that. I added 12 recent original articles relevant to the manuscript.
I changed the figure 1.
As I said I added 12 more original articles relevant to the manuscript.
Reference 9 is related to line 98, reference 2 is related to line 101-108, reference 9 was used here by mistake.
I used reference 7 about the abortion in mice because the involvement of IL 17 in abortion in mice has a similar mechanism of involvement of IL 17 in embryo implantation in endometriosis in humans.
I added “ immunology “ to the chapter title. Anatomy is useful because the main etiopathogenesis of endometriosis, which everyone agrees on, is menstrual reflux, the cells of the immune system are also involved but which then undergo changes.
Also we used MDPI English editing.
Reviewer 2 Report
Comments and Suggestions for Authors
Dear Authors,
I enjoyed reading your paper on pro- and anti- inflammatory cytokines related to endometriosis. I think it could be a useful tool for medical doctors and scientists and could inspire them to new research.
However, some points should get fixed throughout the text.
You should revise the sections of the article and it would be useful for the reader to have the text divided into subsections. This would make the manuscript easier to read and enjoyable.
Figure 1: I suggest you put the letters into the images and report the letters in the legend, or think about another way to fix the legend.
Table 1: please revise the table as it is not clear with this format how it should be read and interpreted.
Section future treatment: why do you focus initially on endometriosis related cancer? Wouldn't you mind more about treatment of endometriosis? Moreover, I'd use a less straight vocabulary in this section as most of the treatments are just proposal or under investigation. Finally, this chapter should be enriched and more detailed.
Comments on the Quality of English LanguageAn English revision by a native is necessary. Some pieces of the text are difficult to read and understand.
Author Response
Cover letter 2
Thank you very much. I highlighted the aim of the manuscript at the end of introduction. I added to the anatomy chapter some immunological changes that is part of etiopathogenic mechanism of endometriosis. According to the aim of our manuscript in proinflammatory, anti-inflammatory, after introduction , anatomy and immunology. Then I added malignant transformation of endometriosis because the aim of our review is to find new treatments, because there is no cure for this disease, so it is linked to the next chapter of future treatment, followed by discussions and conclusion. In the chapter of malignant transformation I added: All this effort of the researchers in this direction is due to the lack of non-invasive diagnosis that postpones obtaining the correct diagnosis for even ten years. In addition the conventional treatment resulted in the amelioration of the disease but without a curative effect.[line 296-299]
I changed the figure.
The table legend:The roles of anti-inflammatory and pro-inflammatory cytokines, their efficacy, mechanism of action, clinical site of action and mode of action, because all this helps us to find out how we can act with the treatment depending on what we are interested in: getting pregnant, relieving pain.
You are right, but as I specified in the aim of the manuscript, researchers are focused on new treatments that have a curative effect, while the known ones we already know, only alleviate the disease. Hence their focus on these treatments starting from the pro-inflammatory and anti-inflammatory cytokines that have gave results in cancers, the two disease having a similar behavior. The studies are under investigation.[line 368-373]
Treatment of endometriosis is widely presented in ESHRE Guidelines of endometriosis. It is subdivided into the treatment of pain, of infertility and surgical treatment. We only do a brief recapitulation, the purpose of the manuscript being to find other new types of targeted treatment. We mention here the analgetics, combined hormonal treatment and contraceptives, progesterone, GnRh agonists, GnRh antagonists, aromatase inhibitors). Infertility treatment consists in ovarian suppression, hormone therapies associated to surgery . Also the most important is surgery: ablation, excision of endometriotic lesions, surgical interruption of pelvic nerve pathways, surgery for deep endometriosis, hysterectomy for endometriosis associated pain. [eshre endometriosis guideline][line 358-366]

Reviewer 3 Report
Comments and Suggestions for Authors
1. Major remarks:
a) the title: "Endometriosis and the Role of Pro-inflammatory and Anti-inflammatory Cytokines, a Narrative Review of the Literature" - the role of what?
b) low number of citations for a review (total 35);
c) the authors jump over issues in nearby sentences and in the main body the statements needs more comment or example;
d) lines 61-62: "The aim was to determine the optimal cut-off point for serum cytokines to differentiate between patterns of ovarian malignancy and endometriosis." - give the cut-off point, please;
e) lines 81-82: "The lesions are classified based on colors: white, red, brown, and black, with the latter being due to the presence of glandular content and adjacent stromal reaction rather than the severity of the disease [1]." - give more explanation (who use this classification? is it important?);
f) the quality and idea of Figure 1 is dim;
g) insufficient information on epigenetic changes and inflammatory cells infiltration (types of lymphocytes, plasmocytes, mastocytes and other).
2. Minor remarks:
a) lines 30 and 469: "pro" - add "-", plese => pro- and anti-inflammatory;
b) line 351: SKA - name first, then abbreviation
Author Response
Cover letter 3
Major remarks
- The role in pathophysiology, added in in the title.
- Thank you, I added 12 original articles.
- I added more comments in the main body of the manuscript and I highlighted the aim of our review at the end of the introduction.
- I added: But the cut-off values for these cytokines in serum were 5.3 pg/ml(IL-6),56.2 pg/ml(IL-8) and 12.56pg/ml(IL-10).[line 66-68]
- I highlighted here the importance of the lesions colour: the following colors: white, red, brown, and black, with the latter being due to the presence of glandular content and adjacent stromal reactions rather than the severity of the disease , important detail for histopathologist. Additionally, hormonal factors play a role, as elevated estrogen levels lead to bleeding in ectopic lesions, with the secondary release of pro-inflammatory cytokines that contribute to iron overload. This results in the infiltration of monocytes and macrophages, which stimulate lipid peroxidation and the accumulation of malondialdehyde (MDA) in the stroma [line 85-94]
- I changed the figure.
- I added to the chapter with anatomy some immunological changes that occur in endometriosis: Mast cells are the key player in allergic response, but also implicated in angiogenesis, fibrosis and pain in endometriosis. The influence of estrogen on mast cell function is a potential factor in pathophysiology of allergic and chronic inflammatory disease. The Mc Callion study shows that endometriotic lesions had significantly higher level of stem cell factors, growth factors and mast cell expression. Mast cells are implicated in increased production of proinflammatory and chemotactic cytokines and also in oxidative stress. IL-6 and IL-8 were increased[X3]. Also an aberrant transcriptome of fallopian tube epithelium and microenvironment changes caused by cytokines in tubal fluid are possible cause for tubal endometriosis. In this Nang Qi study 15 pathway were discovered which induce differential regulation of cytokines production in macrophages and T helper cells by IL-17A and IL-17F. Also hypoxia induced upregulated IL-6 and TNF alpha[X5]. High activity of plasma cells were discovered, modified ratio of Th1/Th2 with increasing Th2 ,increased number of mast cells because of significantly higher level of stem cell factor[X3,X5][line 130=145]
Minor remarks
- I added pro-inflammatory as you asked
- SKA is sequential kinetic activation signaling molecule and I added in the text.
Round 2
Reviewer 1 Report
Comments and Suggestions for Authors
In this revised version of manuscript, authors made some improvements however this reviewer has still major concerns. Please see some of my major concerns below.
1. This manuscript do not provide a clear conclusion to support the main object which is mentioned as 'identifying useful biomarkers for endometriosis detection'.
2. In figure 1, there is no clear representation of hormonal regulation. Besides figure legend is missing.
3. Some phrases that causes major confusion needs to be clarified;
-Line 165: PGE2 is described as an cytokine. Authors better provide a reference that PGE2 is classified as an cytokine or rewrite the sentence to clarify what is actually meant.
-Line 215: Its written that "Cytokines are proinflammatory molecules found in follicular fluid." Please clarify this sentence.
4. Discussion section is written as an introduction. Repeating the literature knowledge. Better be designed as a discussion.
5. In general, throughout the manuscript there is too much repeated knowledge that makes paper difficult to follow.
6. Instead of citing an another review that refers a 1992 paper, authors better use a updated reference to provide information about endometriosis prevalence (line 47). -this was suggested in the previous comments-
This reviewer can not suggest the revised version of manuscript for publication.
Author Response
Cover letter
- I added a text about biomarkers: Also new cytokines biomarkers have been studied starting from the fact that non-invasive diagnosis is missing, the results being promising but more studies are needed Line 597
- I changed the title of the figure to be more consistent, I added a legend. Legend: Infiltration of macrophages, myeloid, mast cells during bleeding secondary release pro-inflammatory cytokines, especially mast cells. IL-37 and IL-17 are primarily produced by T-helper cells. Neutrophils leukocytes contributes to an inflammatory environment. Modified ratio of Th1/Th2 with increasing Th2 was found.
- I tried to clarifie some phrases:
Line 163: Alongside IL-1, another pro-inflammatory cytokine observed in endometriosis is macrophage migration inhibitory factor (MIF).Also prostaglandin PGE2 is involved.
Line 205: Cytokines are found in intrafollicular fluid (Table 1)
- I tried to follow your advice regarding discussions: Manny biomarkers have been studied first in cancers and then in endometriosis. Somme cytokines show increased levels in both pathologies [5].They are studied in order to identify non-invasive markers.
As we have shown, cytokines have various mechanism o action with involvement in embryonic implantation, pain, fibrosis, and depending on what symptoms the patient presents, we focus on relevant biomarkers (table1). New biomarkers panels for disease detection is needed due to its non-specific symptoms and lack of non- invasive diagnosis. As we can see in the table 1, we can identify biomarkers, for example IL-10, whose mechanism of action focuses on angiogenesis and adhesion, which differentiate endometriosis from other pathologies [27].
In other cases proteomics whose expression increases under the influence of interleukins 3 and 4, such as TCF21 can be used either as a biomarker or as a targeted treatment[31].The presence of IL-R1 in all stages of endometriosis, including the early stage, makes this receptor a possible biomarker or targeted treatment.[1].
The anti-inflammatory cytokines, the level of which has been found to increase in advanced stages of the disease, seems to have a protective role in endometriosis. They have a suppressive role in cell proliferation[2].They can be proposed as a treatment due to their inhibitory effect. Line 510-527
The lack of non-invasive diagnosis delay the correct diagnosis for many years. Endometriosis shares a similar behaviour with cancers, even in terms of pathogenesis.This is the reason why the researchers initiative to study cancer biomarkers along with targeted treatments in endometriosis seems reasonable.As I described in chapter 5, additional studies are still needed for a correct conclusion.
Regarding the treatment, considering that there is no curative therapy, the purpose of this review is to present new types of targeted treatments that could have a curative effect. As we presented in chapter 6, the targeted treatments still need in depth studies aand most of them have been studied and are used in other pathologies, especially cancers.Line 543-551.
6.I changed the reference. Line 47 is different :Endometriosis is a pathology characterized by ectopic endometrial tissue implanted and developed on host tissues.[1].
About the prevalence I used ref [41] disease with a prevalence of 5-10% of women of reproductive age. Line 503.
Thank you very much.
Reviewer 3 Report
Comments and Suggestions for Authors
"Figure 1. Schematic representation of the hormonal regulation of endometrial immune cells." - there is no hormones nor pro- and anti-inflammatory factors
Comments on the Quality of English Language
Lines 134-135: "The Mc Callion study ... ." => The McCallion et al. (2022) study .... [11]."
Line 140: "tubal endometriosis. In this Hang Qi study 15 pathway were discovered ..." => tubal endometriosis [12]. In this study 15 pathway were discovered ... OR tubal endometriosis [12]. In the Hang Qi et al. (2020) study 15 pathway were discovered ...
Line 257: "remain unclear.Amalesh Nanda in.." => remain unclear. Amalesh Nanda et al. (2020) in
Line 261: "Extracellular vesicle(EV)associated" => xtracellular vesicle (EV) associated
Line 266: "VEGF-C is upregulated by IL-! Beta and TNF alphaExtracellular vesicle may..." => VEGF-C is upregulated by IL-1 beta and TNF-alpha. Extracellular vesicle may...
Line 271: "SNPs( single nucleotide polymorphism)..." => SNPs (single nucleotide polymorphism)...
line 290: "studies like led by Tamara Knife that showed.." => studies like led by Tamara Knife that showed
Line 299: "endometriosis.IL-13 is lower in endometriosis group according to H. Jorgensen study, but..." => endometriosis. IL-13 is lower in endometriosis group according to H. Jorgensen et al. (2017) study, but.
Line 354: "cytokines,their efficacy, ..." =>cytokines, their efficacy, ...
Line 515: "against IL-1 receptor associated Kinase 4 IRAK4..." => against IL-1 receptor associated kinase 4 (IRAK4)...
Author Response
Cover letter
I changed the title of the figure to be more consistent.
I made all the re quired changes.
Thank you very much.

Round 3
Reviewer 1 Report
Comments and Suggestions for Authors
In this second revised version the manuscript, even though Oala et al. had tried to improve the manuscript, however according to this reviewer, overall quality of the manuscript is not enough to recommend for publication.
This manuscript is still not well organized to present the current literature in a reader friendly way. Overall, manuscript is presented as a copy-paste of the literature without making logical connections between each information. Still, while one paragraph mentions a certain topic, next paragraph has completely different topic with no connection between paragraphs at all.
This approach also causes to may unnecessary repeats of certain information throughout the manuscript.
To avoid this, this reviewer can suggest re-writing all manuscript over, by organizing each cytokine as a individual subdivision and providing all current data relevant to that spesific cytokine under that subdivision designed for each cytokine.
Besides all mentioned recommendations, this reviewer has major concern of the accuracy of the data provided in this manuscript such as:
In Line 176-177, authors mentioned that Boa Weisheng study showed that IL-17 is decreased in endometriosis and IL-6-7-18 and 12 is increased.
This is an example of many un-acceptable mistakes; scientifically misleading data, authors had done so far. In the cited study, Weisheng et al does not report decreased IL-17 in endometriosis. They mention that 17 cytokines out of 38 that they examined, has decreased expression.
This reviewer do not recommend this manuscript for publication.